# Effectiveness of interactive dashboards as audit and feedback tools in primary care: A systematic review

Florence L. Meier⊕, Levy Jäger⊕⊕, Oliver Senn⊕, Stefan Markun, Jakob M. Burgstaller⊕*

Institute of Primary Care, University Hospital Zurich, University of Zurich, Zurich, Switzerland

⊕ These authors contributed equally to this work.
* jakob.burgstaller@usz.ch

## Abstract

Interactive audit and feedback dashboards, which summarize performance using quality indicators, are increasingly used to enhance care processes and outcomes, but their effectiveness in primary care remains underexplored. This systematic review aimed to evaluate the impact of these dashboards in primary care settings by analyzing studies that compared their use to usual care or similar interventions without dashboards. A comprehensive search across MEDLINE (via Ovid), Embase, Cochrane Library, Scopus, and Web of Science through November 2024 was conducted. Risk of bias was assessed using Cochrane tools, and evidence quality was evaluated with GRADE. Relevant data, including features of the interactive dashboards, were extracted, and findings were synthesized narratively and visualized with forest plots. Six studies met the inclusion criteria, comprising five randomized controlled trials and one non-randomized trial, all with low or moderate risk of bias. Four studies incorporating dashboards into multifaceted interventions showed improvements in at least one primary outcome, while two studies using standalone dashboards reported mixed results. Significant heterogeneity in dashboard design, study settings, targeted health conditions, and quality indicators limits the generalizability of these findings. Nonetheless, the results highlight the potential of interactive dashboards to improve quality indicators performance in primary care, particularly as part of broader intervention strategies. Standardized evaluation frameworks and rigorous, consistent reporting are needed in future research to better isolate the effects of interactive dashboards and enhance the robustness and applicability of the evidence in varied primary care contexts.

## Registration

Prospero (CRD42024506727).

**Data availability statement:** All relevant data used for the study has been included in the paper and its Supporting Information files.

**Funding:** The author(s) received no specific funding for this work.

**Competing interests:** At time of writing, the Institute of Primary Care (JMB and SM specifically) is developing an i-A&F dashboard for Swiss general practitioners. As of the start of this review, no funding has been received for this project.

## Introduction

Dashboards are digital tools or interfaces that integrate multiple datasets, data structures, and sources into a single platform, facilitating the visualization and understanding of complex information. In a medical context, dashboards are designed to improve care processes or outcomes of care and can be categorized according to their intended level of implementation [1].

At the micro level, clinical dashboards aggregate data from individual patients, assisting healthcare professionals in monitoring patient-specific data such as vital signs or laboratory values. At the macro level, public health dashboards collect information from entire populations, such as infection rates, to monitor broader public health trends. Between these extremes is the meso level, which pertains to individual healthcare organizations, such as hospitals or practices. Meso-level dashboards typically aggregate data on structures, resources, and the quality of care across multiple patients, with the goal of supporting management and quality improvement initiatives.

Audit and feedback (A&F) is a widely used quality improvement strategy that aims to improve healthcare practice by providing healthcare professionals with feedback summaries on their performance over a specified period, either implemented independently or as an essential part of multifaceted interventions [2–6]. A&F is based on the idea that healthcare professionals will change their behavior to meet desired targets if given feedback demonstrating target deviation of their current performance.

In recent years, the focus has shifted towards electronic A&F (e-A&F), leveraging advancements in health information technology and the widespread adoption of electronic health records (EHRs) [7,8]. This electronic approach often utilizes interactive computer interfaces, also known as interactive dashboards, that present performance summaries in an accessible, engaging format [7]. Quality indicators (QIs)—specific, measurable elements of healthcare—are typically used as standardized metrics within these A&F processes to identify areas needing improvement [7,9,10]. In this context, we define interactive A&F (i-A&F) dashboards here as tools used to aggregate and present QIs in a way that supports e-A&F process.

In particular, healthcare organizations are increasingly encouraged to adopt i-A&F dashboards, and consequently there is a need for rigorous evaluation of the effectiveness of these i-A&F dashboards in achieving their intended goals, especially with regard to improving processes of care and patient outcomes [7,11]. To date, the effectiveness of i-A&F dashboards in the hospital setting has been evaluated with respect to various outcomes [12,13]. However, evidence on the effectiveness of i-A&F dashboards in primary care is limited [11,14,15].

With this systematic review, we aim to assess the effectiveness of interactive dashboards as an A&F tool to improve patient care in primary care.

## Materials and methods

This systematic review followed the guidelines of the "Preferred Reporting Items for Systematic Reviews and Meta-Analyses" (PRISMA) [16] (see S1 Table) and is registered at Prospero (CRD42024506727).

## Data sources and search strategy

An experienced librarian (J.H.) at Careum Bibliothek of the University of Zurich (Zurich, Switzerland) conducted a systematic literature search and included the following databases: MEDLINE® (via Ovid), Embase®, Cochrane Library, Scopus, and Web of Science, from inception to November 4, 2024. The following search terms and their variations were used: "general practitioners", "family practice", "general practice", "primary care", "primary health care", "healthservice", "family physician", "general physician", "general practitioner", "general clinician", "dashboard", "computer interface", "data visualization", "clinical decision support system", "computer graphics", "electronic health record", "electronic patient record", "computerized", "medical record", "healthdata", "interactive", "web portal", "audit", and "feedback". The detailed search strategies can be found in S2 Table. No restrictions were applied for language. Further, reference lists of included studies and reviews were examined to identify additional studies.

## Definition of interactive A&F dashboards

We defined i-A&F dashboards as data management tools designed to enable users to intuitively visualize achievement rates of QIs ("performance summaries") in easily interpretable formats such as graphs, tables, or lists. They should include interactive features that allow users to engage with elements like filtering data, delving into details, or adjusting visualization options to gain deeper insights.

## Eligibility criteria

**Types of studies.** We included randomized controlled trials (RCTs) and non-randomized controlled trials (non-RCTs, controlled before and after studies, controlled trials using non-random methods, controlled interrupted time series studies) evaluating the effectiveness of i-A&F dashboards and excluded conference abstracts and study protocols.

**Participants and setting.** Studies including i-A&F dashboards primarily for individual physicians in a primary care or general practice setting were included. Studies including i-A&F dashboards only used by other medical staff and/or patients as well as i-A&F dashboards used in dentistry were excluded.

**Intervention and comparators.** Any intervention using summaries of QIs displayed in an i-A&F dashboard were included. We examined studies that assessed the effectiveness of i-A&F dashboards compared as intervention to any type of control, including usual care, no intervention, or similar intervention without A&F dashboard use.

**Types of outcome measures.** Outcome measures are QIs and all reported changes in QI achievement rates were considered. All QIs reproduced at least once in another included study were considered for further analysis.

## Study selection and data extraction

Two reviewers (FLM and JMB) independently performed the initial title and abstract screening of references for relevance. This was followed by a full-text analysis to assess eligibility. Disagreements were discussed and resolved by consensus or with third party arbitration (SM). Cohen's Kappa was used to assess the agreement between the raters.

The first reviewer (FLM) extracted relevant data for each included trial into a Microsoft Excel (2016) spreadsheet (Microsoft Corporation, Redmond, WA, USA), which included author, country, year, study design, setting, population, aim, duration, intervention, outcome measures and results, and specific features of A&F dashboard used (e.g., design patterns). All entered data were subsequently verified against the original trials by a second reviewer (JMB).

## Quality assessment

Two reviewers (FLM and JMB) independently assessed the risk of bias using the Cochrane risk-of bias tool for randomized and cluster-randomized trials (RoB 2) [17] and the ROBINS-I [18] tool for non-randomized trials. Since we aimed to assess the effectiveness of i-A&F dashboards to improve achievement rates of QIs, the population considered for the risk

of bias assessment deviated from the standard templates (see also "Participants and Setting"). Therefore, we defined the physicians (users) as "participants" in the RoB 2 tool for questions 2.1a (cluster) and 2.1 (randomized) and the research team conducting the study and providing the A&F dashboard as "carers and people delivering the interventions" for question 2.2. "Assessors" were defined as the analysts of the data in questions 4.3a (cluster) and 4.3 (randomized) and in the ROBINS-I tool for question 6.2.

To assess the overall quality of evidence, we used the Grading of Recommendations Assessment, Development, and Evaluation (GRADE) tool [19] for the studies included in the further analysis (see Data Synthesis and Analysis).

### Assessment of dashboard features and dashboard evaluation

To systematically evaluate the features of the included i-A&F dashboards, we adopted a structured approach to dashboard design and evaluated the i-A&F dashboards according to a suggested framework. We extracted the information either from the main text or from tables or figures, either in the main manuscripts or appendices, or from previous publications such as protocols or pilot trials, or subsequent publications such as process evaluations or assessments.

### Dashboard design patterns

Bach et al. [20] proposed design patterns providing a structured approach to developing more effective and user-friendly dashboards. They grouped these patterns into two high-level groups: *Dashboard Content*, focusing on what information is shown on the dashboard, including data information, meta information, and visual representation, and *Dashboard Composition*, addressing how components are laid out and structured across one or many dashboard pages, including page layout, screenspace, structure, interaction, and color usage. Detailed information can be found in S3 Table.

### Evaluation of dashboards

Zhuang et al. [21] introduced a comprehensive evaluation framework for dashboards used in healthcare, grounded in a systematic review of existing literature. This framework consists of seven evaluation scenarios, defined as "an umbrella term for all methods and objectives that address a single, particular aspect of the interaction between user and dashboard". There are categorized into three main themes: *Interaction Effectiveness* (with the evaluation scenarios Task Performance, Behaviour Change and Interaction Workflow), *User Experience* (Perceived Engagement and Potential Utility), and *System Efficacy* (Algorithm Performance and System Implementation). Detailed information can be found in S4 Table.

### Data synthesis and analysis

Due to the large heterogeneity of study designs, interventions, populations, and outcome measures between the studies, the results of a meta-analysis aggregating the various effect sizes across studies would be difficult to interpret, and we refrained from conducting a meta-analysis. Instead, we conducted a narrative synthesis to explore variations in study and dashboard designs, populations, and QIs. The effect sizes found in interventions for QIs replicated in at least one other included study are summarized as odds ratios (ORs) with 95% confidence intervals (CIs) and presented in forest plots (further analysis). Intervention effects reported as proportions, changes in percentage, or relative risk ratios (RRs) were converted to corresponding ORs. Further, the forest plots only include crude ORs, regardless of whether adjusted measures were provided in the original studies, to facilitate between-study comparison.

### Results

Our search strategy initially retrieved 9,354 articles across five databases is shown in Fig 1. After deduplication, 5,096 unique articles remained. From these, we identified 76 articles for full-text review, resulting in five articles being included in this review. Additional citation screening identified nine more articles for full-text review, of which one were included.

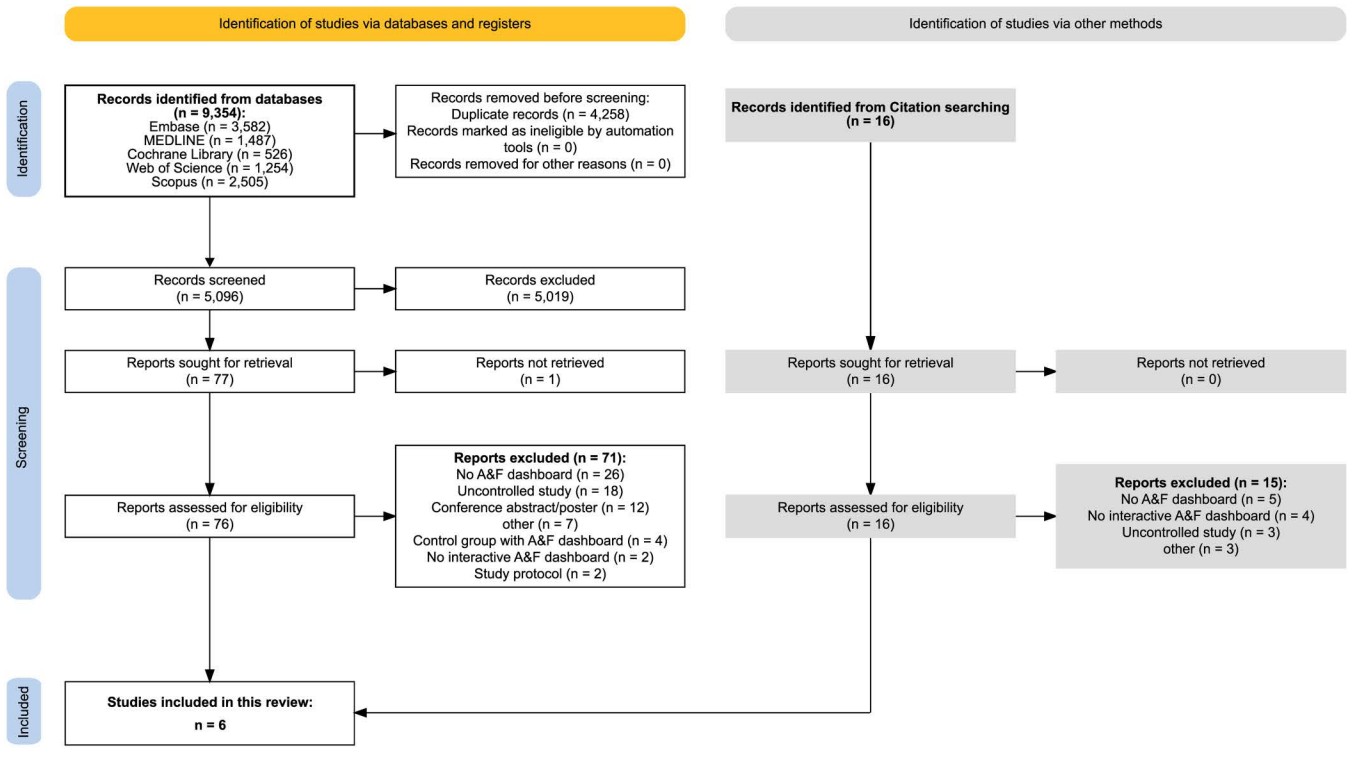

**Fig 1. Flowchart for study selection.**

Finally, six studies were included in our systematic review. The reviewers (FLM and JMB) achieved interrater agreement in the screening of titles/abstracts and full texts, with respective Kappa scores of 0.84 and 0.85. Reasons for exclusion after full-text screening are detailed in S5 Table.

## Overview and characteristics of included studies

Table 1 shows the characteristics of the included studies. Of the six included studies, five were RCTs: three cluster RCTs [22–24] and two stepped wedge cluster RCTs [25,26]. The remaining study was a non-randomized controlled trial (quasi experimental trial) [27]. In the two stepped wedge cluster RCTs [25,26], baseline numbers from the pre-intervention period served as comparators, while in all other studies, the comparator was usual care. The publication years ranged from 2010 to 2023.

One studie took place in primary care centers [24], the remaining in primary care/general practices [22,23,25–27]. One study focused on prescribing of antibiotics [23] and one focused on high-risk prescribing of nonsteroidal anti-inflammatory drugs (NSAIDs) or selected antiplatelet agents [25]. The remaining four studies focused on cardio-metabolic and -vascular diseases [22,24,26,27]. In three studies, the patient population was general practice patients [23,24,26], two studies included type 2 diabetes (T2D) patients [22,27], and one study included patients that were particularly vulnerable to adverse drug events related to NSAIDs or antiplatelet agents [25]. Further, four studies added additional elements ("multifaceted intervention") to their intervention besides the i-A&F dashboard (e.g., clinical staff training, recommendations, educational materials) [24–27]. The duration of the interventions ranged from 16 weeks to 17.5 months (Table 1).

Two studies [22,23] declared the number of participating physicians (160 and 573, respectively) and all studies [22–27] declared the number of practices or sites included (range 12–76). Details about the primary and secondary outcomes of the studies are provided in Table 2.

**Table 1. Study characteristics.**

| Author, Year, Country | Study design | Setting | Study aim | Patient population | Intervention | Control | Multifaceted intervention | Duration of interventions |
|---|---|---|---|---|---|---|---|---|
| de Lusignan, 2021, United Kingdom [27] | quasi-experimental study | primary care practices | prescribing of metformin and aspirin (in patients with type 2 diabetes) | type 2 diabetes patients visited primary care practice | Dashboard (graphical presentations) for either metformin (group A) or aspirin (group B) prescription; recommendations for prescription | usual care (either group B or A, "cross over") | yes | 6 months |
| Dreischulte, 2016, Scotland [25] | cluster-randomized stepped wedge trial | primary care practices | high risk prescribing of NSAIDs or selected antiplatelet agents | general practice patients with one or more risk factors that made them particularly vulnerable to adverse drug events related to NSAIDs or antiplatelet agents | professional education and outreach visit by pharmacist; written material and newsletter; financial incentives; informatics tool (web based) graphically displaying feedback | baseline | yes | 48 weeks |
| Guldberg, 2011, Denmark [22] | cluster randomized controlled trial | primary care practices | quality of type 2 diabetes care | type 2 diabetes patients visited primary care practice | electronic feedback system (graphical presentations incl. peer comparisons) | usual care | no | 15 months |
| Jones, 2023, Australia [26] | cluster-randomized stepped wedge trial | primary care practices | management of chronic vascular diseases (in patients with chronic kidney disease, type 2 diabetes, or cardiovascular disease) | general practice patients (based on Royal Australian College of General Practitioners [RACGP] recommendations: age ≥18 years and being an active patient (attendance at the same general practice ≥three times in the past 24 months)) | 1) electronic-technology tool (generating graphs and lists for audit) 2) education provided to practices 3) monitoring and support provided to practices | baseline | yes | 16 weeks |
| Linder, 2010, USA [23] | cluster randomized controlled trial | primary care practices | prescribing of antibiotics for acute respiratory infections (ARI) | general practice patients visited primary care practice | ARI Quality Dashboard (graphical presentations) | usual care | no | 9 months |
| Peiris, 2015, Australia [24] | cluster-randomized controlled trial, parallel arm | primary health-care centers | measuring of cardiovascular disease risk factors (guideline indicated) and medication prescribing for high risk patients (guideline indicated) | general practice patients (based on Australian guideline vascular risk screening recommendations: all Aboriginal and Torres Strait Islander people ≥35 years and all others ≥45 years (no upper age limit) who had attended the service ≥3× in the previous 24-month period and at least once in the previous 6-month period.) | 1) computerized decision support tool 2) audit and feed-back tool (graphical presentations) 3) staff training | usual care | yes | ~17.5 months* |

**Table 2.** Summary of results.

| Author, Year | Sites/ Physicians/ Patients, n | Primary Target(s) | Primary Outcome(s) Results | Secondary Target(s) | Secondary Outcome(s) Results |
|---|---|---|---|---|---|
| de Lusignan, 2021 [27] | 12/ n.r./ 5644 | 1) Reduction in inappropriate metformin prescribing in people with T2D with reduced renal function. 2) Reduction in inappropriate aspirin prescribing in people with T2D without CVD or CKD and in those aged ≥70 years. | 1) Inappropriate metformin prescribing: no regression analysis possible (very low numbers of people) 2) Inappropriate aspirin prescribing: OR 0.44, 95% CIs: 0.27 to 0.72 | n.r. | n.r. |
| Dreischulte, 2016 [25] | 33/ n.r./ 33060 | Reduction of patient-level exposure to any of nine measures of high-risk prescribing of NSAIDs or selected antiplatelet agents. | Rate of high-risk prescriptions: aOR 0.63, 95% CIs: 0.57 to 0.68 | 1) Ongoing (prescribed within the previous year) and new (not prescribed within the previous year) high-risk prescribing. 2) Rates of the nine prescribing outcome measures individually. 3) Emergency hospital admissions for a) gastrointestinal bleeding, b) acute kidney injury, or c) heart failure. | 1) Ongoing High-Risk Prescribing: aOR 0.60; 95% CIs: 0.53 to 0.67 New High-Risk Prescribing: aOR 0.77; 95% CIs: 0.68 to 0.87 2) Significant reductions were observed across eight out of the nine high-risk prescribing measures, with adjusted odds ratios ranging from 0.27 to 0.78. The only exception was NSAID prescribing in patients with heart failure, where no significant change was found. 3) Hospital admissions for a) gastrointestinal bleeding: RR 0.66, 95% CIs: 0.51 to 0.86 b) acute kidney injury: RR 0.84; 95% CIs: 0.68 to 1.09 c) heart failure: RR 0.73; 95% CIs: 0.56 to 0.95). |
| Guldberg, 2011 [22] | 76/ 160/ 2458 | Processes of care according to the Danish evidence-based diabetes guidelines for general practitioners on 1) prescriptions redeemed for T2D treatments a) oral antidiabetic agents, b) insulin, c) lipid-lowering medication, and d) blood pressure medication 2) measuring of HbA1c 3) measuring of serum cholesterol 4) visits to ophthalmologists | Rates of redeemed prescriptions: 1a) Oral antidiabetic medication: TD 20.9%, 95% CIs: 7.9 to 34.8 1b) Insulin: TD 21.4%, 95% CIs: 9.9 to 32.8 1c) Lipid-lowering medication: TD 19.7%, 95% CIs: 6.1 to 33.2 1d) Blood pressure medication: TD 11.3%, 95% CIs: 1.4 to 21.2 2) Annual HbA1c measurements: TD –6.4%, 95% CIs: –17.4 to 4.6 3) Cholesterol measurements: -sustained treatment: TD 6.7%, 95% CIs: –0.6 to 14.0 -initiated treatment: TD 14.7%, 95% CIs: –2.4 to 31.9 4) Ophthalmologist visits: TD 1.2%, 95% CIs: –2.1 to 4.5 | Mean changes in levels of 1) HbA1c 2) serum cholesterol. | 1) HbA1c levels: TD –0.05; 95% CIs: –4.2 to 14.2 2) serum cholesterol levels: TD 0.00; 95% CIs: –2.1 to 8.1 |

*(Continued)*

| Author, Year | Sites/ Physicians/ Patients, n | Primary Target(s) | Primary Outcome(s) Results | Secondary Target(s) | Secondary Outcome(s) Results |
|---|---|---|---|---|---|
| Jones, 2023 [26] | 8/ n.r./ 37946 | Multiple prespecified outcomes assessing risk factor assessment, risk factor presence, diagnostic testing in those at risk, chronic disease presence and chronic disease man-agement (in total 45, no single prespecified primary outcome was selected). | -Kidney health checks (eGFR, uACR and BP) in those at risk: OR 1.34; 95% CIs: 1.26 to 1.42 -Coded diagnosis of CKD: OR 1.18; 95% CIs: 1.09 to 1.27 -T2D diagnostic testing (fasting glucose or HbA1c) in those at risk: OR 1.15; 95% CIs: 1.08 to 1.23 -uACR in patients with T2D: OR 1.78; 95% CIs: 1.56 to 2.05 -Documented eye checks within recommended frequency in patients with T2D decreased: OR 0.85; 95% CIs: 0.77 to 0.96 '-No significant changes in other assessed variables. | n.r. | n.r. |
| Linder, 2010 [23] | 27/ 573/ 136633 | Intent-to-intervene antibi-otic prescribing rate for all ARI visits. | Antibiotic prescribing for all ARI visits: OR 0.97; 95% CIs: 0.7 to 1.4 | Intent-to-intervene anti-biotic prescribing rate for1) antibiotic-appropriate diagnoses 2) and non–antibiotic-appropriate diagnoses 3) for individual ARI diagnoses. As-used analysis antibiotic pre-scribing rate between intervention clinicians who used the dash-board at least once with intervention clinicians who did not use the dashboard for 4) for all ARI diagnoses 5) antibiotic-appropriate diagnoses 6) non–antibiotic-appropriate diagnoses. | No significant difference in antibiotic prescribing for1) antibiotic–appropriate ARIs 2) non–antibiotic-appropriate ARIs 3) any individual ARI. Dashboard users and non-users in antibiotic prescribing for 4) all ARIs (OR 0.72; 95% CIs: 0.54 to 0.96) 5) non–antibiotic-appropriate ARIs (OR 0.78; 95% CIs: 0.53 to 1.15). 6) non–antibiotic-appropriate ARIs (OR 0.63; 95% CIs: 0.45 to 0.86). |

*(Continued)*

**Table 2.** (Continued)

| Author, Year | Sites/ Physicians/ Patients, n | Primary Target(s) | Primary Outcome(s) Results | Secondary Target(s) | Secondary Outcome(s) Results |
|---|---|---|---|---|---|
| Peiris, 2015 [24] | 60/ n.r./ 38725 | 1) The proportion of eligible patients who received appropriate screening of CVD risk factors by the end of study. 2) The proportion of eligible patients defined at baseline as being at high CVD risk, receiving recommended medication prescriptions at the end of study. | 1) Receiving comprehensive CVD risk factor screening: RR 1.25, 95% CIs: 1.04 to 1.50 2) Prescribing of recommended medications for high-risk patients: RR 1.11; 95% CIs: 0.97 to 1.27 | 1) Measurements of individual CVD risk factors a) smoking status, b) BP, c) lipids, d) BMI, e) eGFR, and f) uACR. 2) Escalation of drug prescription among patients at high CVD risk, either newly prescribed or additional numbers of a) antiplatelet, b) BP-lowering and c) lipid-lowering agents). 3) a) BP and b) serum lipid levels among people at high CVD risk. 4) newly recorded CVD-related diagnoses. | 1) Measurements of individual CVD risk factor: a) smoking status: OR 1.04; 95% CIs: 0.96 to 1.13 b) BP: OR 1.08; 95% CIs: 0.99 to 1.18 c) lipids: OR 1.19; 95% CIs: 1.03 to 1.37 d) BMI: OR 0.97; 95% CIs: 0.77 to 1.23 e) eGFR: OR 1.06; 95% CIs: 0.97 to 1.15 f) uACR: OR 1.23; 95% CIs: 0.84 to 1.80 2) Treatment escalations (new prescriptions or increased numbers) for a) antiplatelet: OR 4.79; 95% CIs: 2.47 to 9.29 b) lipid-lowering: OR 3.22; 95% CIs: 1.77 to 5.88 c) blood pressure-lowering medications: OR 1.89; 95% CIs: 1.09 to 3.26 3a)     Mean systolic BP: TD −0.8 mmHg; 95% CIs: −2.0 to 0.4 3b)     Mean low-density lipoprotein cholesterol: TD −0.05 mmol/L; 95% CIs: −0.12 to 0.01 4) No differences in the proportion with newly recorded CVD diagnoses (P=0.72). |

aOR = adjusted odds ratio; ARI = acute respiratory infections; BMI = body mass index; BP = blood pressure; CIs = confidence intervals; CKD = chronic kidney disease; CVD = cardiovascular disease; eGFR = estimated glomerular filtration rate; OR = odds ratio; NSAIDs = nonsteroidal anti-inflammatory drugs; RD = risk difference; RR = risk ratio; TD = treatment difference; T2D = type 2 diabetes; uACR = urine albumin creatinine rati

Most studies extracted the data they analyzed directly from the EHR [23–26], and the dashboard data was updated between weekly and three times per year. The i-A&F dashboards were accessible via the EHR [23,26], a web-based portal [24,25], or were additional programs [22,27] (S6 Table).

## Quality assessment of included studies

Fig 2 and S7 Table shows the risk of bias of all included studies. One RCT found to have a low risk of bias in all domains [25], two RCTs [24,26] were judged to raise some concerns in one domain, and two RCTs [22,23] in three domains. The non-RCT [27] showed moderate risk of bias in the domains "Bias in measurement of outcomes" and "Bias in selection of the reported results".

## Quality of evidence

The quality of evidence for the studies which reported similar QIs was assessed using an adapted form of GRADE [19] because single estimate of effects were not calculated [28]. The quality of evidence was assessed to be low with the main downgrading factors being methodological limitations of the studies and indirectness (S8 Table).

## Effects of the interventions

Table 2 shows the summary of results for all included studies and includes the number of sites, physicians and participants, description and results of both primary and secondary outcomes in each study.

Four studies using the i-A&F dashboard as a core component within a multifaceted intervention reported a favorable effect of at least one of their primary outcomes. In patients with T2D, de Lusignan et al. [27] found that practices with dashboard access significantly lowered their inappropriate aspirin prescribing (OR 0.44, 95% CIs: 0.27 to 0.72). In another study, high-risk prescribing of nonsteroidal anti-inflammatory drugs (NSAIDs) or selected antiplatelet agents was significantly reduced in the intervention group (adjusted OR 0.63, 95% CIs: 0.57 to 0.68) [25]. Jones et al. [26] reported in 7 of 45 QIs a significant effect in the intervention group for kidney health checks (estimated glomerular filtration rate (eGFR), urine albumin creatinine ratio (uACR) and blood pressure) (OR 1.34, 95% CIs: 1.26 to 1.42) and T2D diagnostic testing (fasting glucose or HbA1c) (OR 1.15, 95% CIs: 1.08 to 1.23) in those patients at risk for cardiovascular disease (CVD);

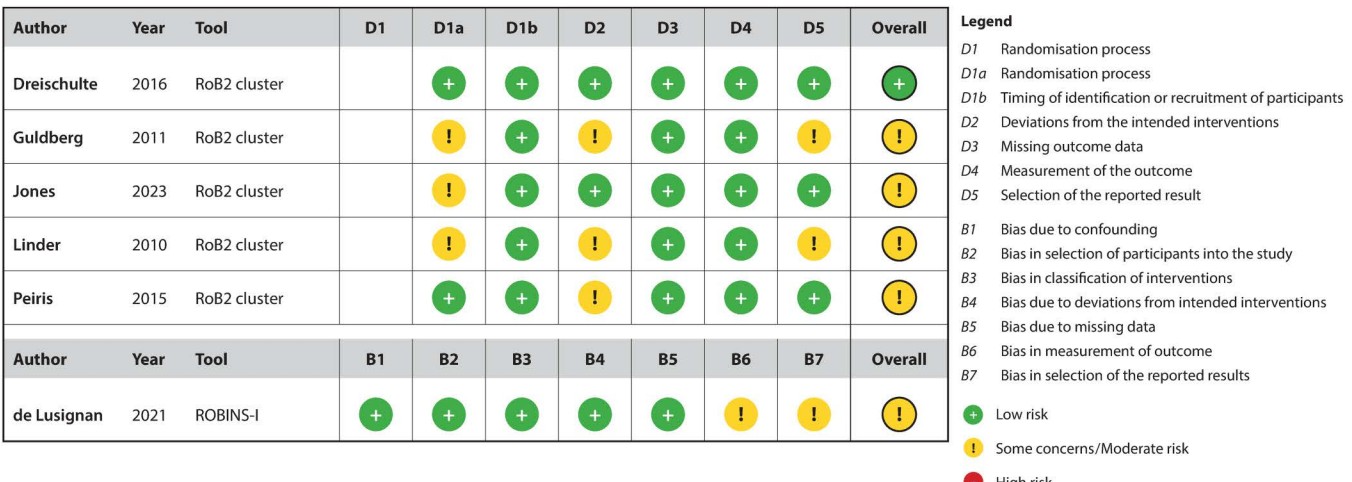

**Fig 2. Risk of bias of included studies.**

coded diagnosis of chronic kidney disease (CKD) (OR 1.18, 95% CIs: 1.09 to 1.27); uACR (OR 1.78, 95% CIs: 1.56 to 2.05) and reduced documented eye checks (OR 0.85, 95% CIs: 0.77 to 0.96) in patients with T2D. In the study of Peiris et al. [24], the intervention was associated with improved overall risk factor measurements (RR 1.25, 95% CIs: 1.04 to 1.50).

The remaining two studies used the i-A&F dashboard as a single-component intervention. Guldberg et al. [22] reported favorable outcomes in the intervention group for redeemed recommended prescriptions during follow-up for oral antidiabetic treatment (treatment difference (TD) 20.9%, 95% CIs: 7.9 to 34.8), insulin treatment (TD 21.4%, 95% CIs: 9.9 to 32.8), lipid-lowering medication (TD 19.7%, 95% CIs: 6.1 to 33.2), and blood pressure medication (TD 11.3%, 95% Cis: 1.4 to 21.2) in patients with T2D. On the other hand, Linder et al. [23] could not report a beneficial effect for their main outcome, antibiotic prescribing for all acute respiratory infections (ARIs) (OR 0.97; 95% CIs: 0.7 to 1.4). However, in further analyses they found that GPs in the intervention group who used the i-A&F dashboard were less likely to prescribe antibiotics for all ARIs (OR 0.72; 95% CIs: 0.54 to 0.96) and for non-antibiotic-appropriate ARIs (OR 0.63; 95% CIs: 0.45 to 0.86) compared with non-dashboard users in the intervention group.

### Further analysis

Fig 3 and S9 Table summarizes the results of interventions on similar QIs involved in at least two studies. Only three studies could be included in this analysis [22,24,26].

Lipid testing (cholesterol measurement), evaluated in all three studies, showed a favorable effect with ORs ranging from 1.56 to 2.08, suggesting a consistent benefit. Similarly, uACR measurement and smoking status, studied by Jones et al. [26] and Peiris et al. [24], had ORs ranging from 1.29 to 1.51 and 1.10 to 1.21, respectively, indicating positive, though modest, effects.

Conversely, five QI categories showed no consistent effect across studies. For example, lipid medication prescriptions and blood pressure medication prescriptions, examined in all three studies, showed mixed results, with Jones et al. [26] reporting mostly small and non-significant effects. HbA1c testing [22,26] showed a tendency toward a less favorable effect for the intervention. For eye examinations [22,26] and eGFR measurement [24,26], the studies showed conflicting results.

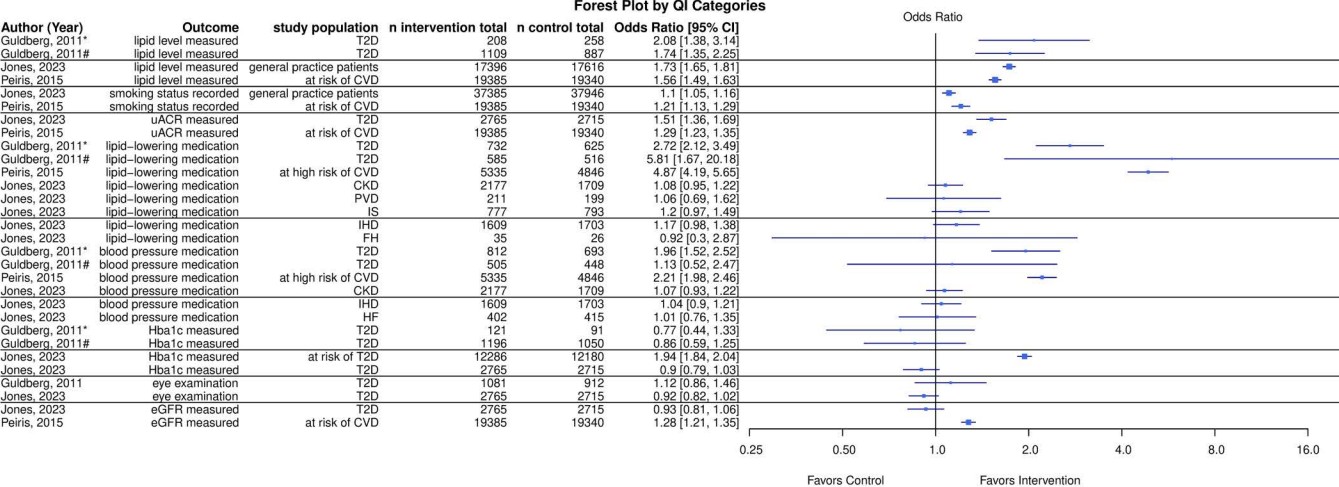

**Fig 3. Effects of interventions.** Effect sizes and confidence intervals of interventions for QIs replicated in at least one other included study, converted to corresponding odds ratios where necessary. Abbreviations: *initiated treatment; #sustained treatment; CKD = chronic kidney disease; CI = confidence interval; CVD = cardiovascular disease; eGFR = estimated glomerular filtration rate; FH = familial hypercholesterolemia; HF = heart failure; IHD = ischemic heart disease; IS = ischemic stroke; PVD = peripheral vascular disease; QI = quality indicator; T2D = type 2 diabetes; uACR = urine albumin creatinine ratio.

## Dashboard usage

Linder et al. [23] stated that 28% of the clinicians in the intervention group used the i-A&F dashboard at least once. Peiris et al. [24] reported only that 90% of the intervention sites uploaded data to the dashboard, but did not further revealed how often the i-A&F dashboard was used (S6 Table).

## Dashboard features and evaluation

The features of the i-A&F dashboards varied considerably between the studies and were rarely described. In most cases, we had to extract the information from screenshots of the dashboards, which made the evaluation and interpretation more difficult.

Regarding the *content* design patterns (S10 Fig, for definitions see S3 Table), all six i-A&F dashboards showed aggregated data ("aggregation"), whereas only three offered complete view of the data ("detailed data set"). Filtering functionalities ("filtering") were present in four i-A&F dashboards, enabling users to refine data views, and thresholds were implemented in three i-A&F dashboards. Meta data elements were inconsistently included across dashboards. Five i-A&F dashboards released information about the last update ("update information"), and only one i-A&F dashboards explained what the data represents ("data description"). Disclaimers were absent across all i-A&F dashboards. In terms of visual representation, the most commonly used formats were detailed visualizations (combined visual components including both graphs and additional informational elements), which were present in all six i-A&F dashboards. Tables were utilized in four i-A&F dashboards. Further visual elements such as gauges and trending arrows were only used to a limited extent.

Regarding the *composition* design pattern (for definitions see S3 Table), different page layouts such as open, table and grouped were used. Most i-A&F dashboards demonstrated varying levels of screenspace utilization. In terms of structural design, single-page layouts dominated, being used in four dashboards. Interaction capabilities varied across dashboards: Drilldown functionality, allowing users to explore specific data subsets, was supported by all i-A&F dashboards. Three i-A&F dashboards allowed to explore data elements and obtain new data views ("exploration"). Further, three i-A&F dashboards enabled moving between pages or sections ("navigation") and only one allowed customization ("personalization"), which supports tailoring the dashboard to user preferences. A detailed overview can be found in S10 Fig.

In the evaluation scenarios (S10 Fig, for definitions see S4 Table), every study demonstrated an improvement in at least one performance metric or task ("task performance"). However, for the four studies [24–27] involving multifaceted interventions, it remained unclear to what extent the dashboards themselves contributed to supporting task completion effectively. Three studies observed positive behavior changes attributed to the use of the i-A&F dashboard ("behavior change"). However, results for integration into daily workflows ("interaction workflow") were mixed, with both positive and negative findings. Two studies discussed functionalities not currently implemented in the i-A&F dashboard but identified as valuable additions to enhance its future potential ("potential utility"). Notably, in Linder et al. [23], this feedback was gathered during the pilot study, and the suggested features were incorporated into the final version of the i-A&F dashboard. Four studies reported that most of the subjective feedback collected on the i-A&F dashboard's usability and user satisfaction ("perceived engagement") was positive. This feedback was gathered through either user involvement during development [22,23] or post-hoc interviews [24,25]. Only two i-A&F dashboards [23,24] were directly integrated into existing EHRs. The other four i-A&F dashboards [22,25–27] were independent of the EHR ("system implementation"). Lastly, as the "algorithm performance" scenario pertains mainly to decision support processes, we excluded it from our analysis.

Fig 4 provides an overview of the total number of studies that used a particular dashboard feature or met a particular evaluation scenario.

## Discussion

This systematic review included six studies evaluating the effectiveness of interactive dashboards as an A&F tool to improve patient care in primary care settings, focusing on areas such as antibiotic prescribing, high-risk medication prescribing, and cardio-metabolic diseases. Four studies assessed i-A&F dashboards as a component of multifaceted

| Data | | Meta Data | | Visual Representation | | Page Layout | | Screenspace | |
|---|---|---|---|---|---|---|---|---|---|
| *Variables* | *No. of Studies* | *Variables* | *No. of Studies* | *Variables* | *No. of Studies* | *Variables* | *No. of Studies* | *Variables* | *No. of Studies* |
| Detailed Data Sets | 3 | Data Source | 2 | Tables | 4 | Open | 2 | Screenfit | 4 |
| Aggregation | 6 | Disclaimer | 0 | Lists | 1 | Table | 1 | Overflow | 1 |
| Filtering | 4 | Data Description | 1 | Detailed Visualization | 6 | Stratified | 0 | Detail on demand | 3 |
| Derived Values | 1 | Update Information | 5 | Miniature Charts | 1 | Grouped | 2 | Parameterization | 4 |
| Thresholds | 3 | Annotations | 1 | Gauge & Process Bars | 0 | Schematic | 0 | Multiple Pages | 3 |
| Single Value | 3 | | | Trend Arrows | 1 | | | | |
| | | | | Pictograms | 1 | | | | |
| | | | | Numbers | 2 | | | | |

| Structure | | Interaction | | Color | | Evaluation | |
|---|---|---|---|---|---|---|---|
| *Variables* | *No. of Studies* | *Variables* | *No. of Studies* | *Variables* | *No. of Studies* | *Variables* | *No. of Studies* |
| Single Page | 4 | Exploration | 3 | Distinct | 2 | Task Performance | 6 |
| Parallel | 1 | Navigation | 3 | Data Encoding | 2 | Behavior Change | 3 |
| Hierarchic | 0 | Personalization | 1 | Semantic | 3 | Interaction Workflow | 3 |
| Open | 0 | Drilldown | 6 | Emotive | 0 | Perceived Engagement | 4 |
| | | | | | | Potential Utility | 2 |
| | | | | | | Algorithm Performance | n.a. |
| | | | | | | System Implementation | 6 |

**Fig 4. Overview of dashboard features and evaluation.** Number of studies that used a particular dashboard feature or met a particular evaluation scenario. Abbreviations: n.a. = not applicable.

interventions, while two evaluated them as standalone interventions. The findings suggest that i-A&F dashboards can positively influence specific QIs, particularly when used as components of multifaceted interventions. However, variability in study designs, interventions, and observed QIs limited the ability to draw generalized conclusions or perform a meta-analysis. Furthermore, significant variation in dashboard design features was found across studies, suggesting another level of effect modification that should be considered in future research on dashboard effectiveness.

## Comparison with existing literature

Several systematic reviews have explored the effectiveness of A&F dashboards across various clinical settings and outcomes. Garzon et al. [14] focused on interactive dashboards to optimize antibiotic prescribing in primary care, including ten studies. The review concluded that interactive dashboards, especially when integrated with other interventions, can help reduce antibiotic prescribing in primary care. Unlike our study, they did not clearly define "interactive" and included data from dental and emergency department settings.

Tuti et al. [7] focused on the effectiveness of e-A&F interventions utilizing interactive computer interfaces and their underlying behavior change theories. They included seven studies covering diverse A&F domains, from drug prescribing and management to disease and care management. Only three studies were conducted in a primary care setting. The review found high heterogeneity in the interventions, making it challenging to determine a reliable average effect.

Van den Bulck et al. [8] reviewed 29 studies assessing the impact of static and interactive e-A&F in primary care, also covering diverse A&F domains. Among the eight studies with interactive e-A&F, five demonstrated a positive impact on healthcare provider performance and outcomes. However, this review used a broader definition of "interactive" and included a broader scope of e-A&F, not limited to interactive dashboards as defined in our study. Additionally, the methods used to assess the intervention's impact on outcomes were not clearly specified.

Xie et al. [15] reviewed eleven studies on the effectiveness of clinical dashboards used as A&F or clinical decision support systems on medication prescription/adherence and test ordering. Their findings suggested limited evidence for dashboards' effectiveness in improving these outcomes. In contrast to our study, they also included hospital-based studies and did not focus solely on interactive A&F dashboards.

Finally, Dowding et al. [11] aimed to assess the impact of clinical and quality dashboards across various healthcare settings, primarily outside primary care. Among the eleven studies included, results were mixed; some dashboards positively impacted patient outcomes and care processes, while others had no observable effect.

## Implementation of interactive dashboards

Four studies included in our review [24–27] used the i-A&F dashboard as a core component within a multifaceted intervention aimed at enhancing healthcare outcomes. This integration complicates the isolation of the dashboard's specific contribution, as the effects of individual components are not assessed independently. Consequently, it becomes challenging to determine whether improvements in healthcare performance can be attributed to the dashboard itself or to other elements of the intervention, or to understand why some dashboard interventions yield positive effects while others do not. Adding to this challenge, only Linder et al. [23] investigated dashboard usage in detail, revealing that merely a quarter of clinicians in the intervention group accessed the dashboard at least once, further complicating the assessment of its impact.

Recent research emphasizes the importance of following frameworks to build dashboards. The framework proposed by Zhuang et al. [21] evaluates interaction effectiveness, user experience, and system efficacy. They emphasize the need for an iterative design process informed by user feedback and evaluation results, highlighting the importance of assessing both the process of dashboard usage (e.g., interaction workflows, perceived engagement) and its outcomes (e.g., task performance, behavior change, potential utility), either during the design phase or through reevaluation after implementation. Three included studies [22,24,25] conducted post-trial process evaluations or assessments where we were able to identify relevant information aligned with the framework by Zhuang et al., albeit not in the detailed manner recommended by the framework. Unfortunately, it remains unclear whether the feedback gathered in these analyses was incorporated into future versions of the i-A&F dashboards. In contrast, Linder et al. [23] conducted a pilot study in which some suggested improvements were successfully integrated into the dashboard used in the trial.

The framework proposed by Bach et al. [20] offers a structured approach for analyzing dashboard composition, focusing on content and structure, and emphasizing the importance of balancing the amount of information presented with ease of access for users. It advocates for avoiding common pitfalls such as excessive data, overly complex charts, lengthy text, visual clutter, inconsistent color schemes, and confusing layouts, which can hinder usability and effectiveness. Determining what constitutes a "good" dashboard, however, depends on various subjective and domain-specific factors. As such, we refrained from making overall judgments about the quality of the included dashboards.

Our evaluation revealed that while certain data patterns, such as aggregation and detailed visualizations, were consistently implemented across i-A&F dashboards, there were significant gaps in meta-data features which could impact user comprehension. However, our analysis was further complicated by the limited access to the dashboards themselves, as we had to rely solely on screenshots.

## Implications for practice and future research

Our findings emphasize the importance of consistently applying design best practices and frameworks to improve user engagement and maximize utility. Actively involving end users—particularly clinicians in our study—through a structured, user-centered approach is crucial for creating dashboards that are both effective and widely adopted. Continuous improvement is essential for ensuring these tools remain effective and user-friendly over time, particularly given the heterogeneity in digital and health literacy among clinicians, which may influence uptake and successful utilization. For example, Linder et al. [23] demonstrated that while only a quarter of clinicians used the dashboard, those who did were significantly less likely to prescribe antibiotics, highlighting the potential impact of well-designed and continuously refined dashboards. Further, future research should also establish consistent reporting standards regarding actual dashboard usage analytics to more accurately assess dashboard implementation and impact.

Several important implementation and contextual factors were not systematically addressed in this review, as they were outside the predefined scope and were infrequently reported in the included studies. These include data security and privacy considerations (e.g., GDPR, HIPAA compliance), data input quality and reliability, digital inclusion and infrastructure readiness in low-resource settings, and details about interoperability with existing health information systems. While these factors are critical for the real-world adoption and effectiveness of interactive dashboards in primary care, they were not the primary focus of our analysis. We recommend that future research and reporting on digital A&F tools in primary care explicitly address these domains to enhance the applicability and generalizability of findings.

### Limitations

The findings of this systematic review are limited due to the inconsistent and variable definitions of the terms "dashboard" and "interactive" across the literature as well the related heterogeneity in the conceptualization and reporting of these tools. This challenge was further compounded by the recent introduction of the MeSH (Medical Subject Headings) term "Dashboard Systems" by the National Library of Medicine in 2024, which limited standardized indexing for PubMed articles at the time of our search. Despite these challenges, we are confident that our sensitive search strategy, coupled with manual screening of potentially eligible studies cited in the included articles and systematic reviews, minimized the likelihood of missing relevant studies. However, our framework evaluations were constrained by limited access to the A&F dashboards included in the studies, underscoring the importance of future research providing full access to dashboard interfaces to facilitate more comprehensive evaluations.

### Conclusions

Our findings suggest that interactive dashboards typically within multifaceted A&F interventions can improve specific quality indicators performance in primary care. However, the generalizability is limited by the heterogeneity in study designs, implementation settings, health conditions, quality indicators, and reporting quality. Frameworks should be used to improve the validity of evidence in future studies, in particular evaluating the independent effect of i-A&F dashboards on healthcare performance.

### Supporting information

**S1 Table. PRISMA checklist.**
(DOCX)

**S2 Tables. Search strategies.**
(DOCX)

**S3 Table. Dashboard design patterns.**
(DOCX)

**S4 Table. Evaluation scenarios.**
(DOCX)

**S5 Table. Excluded studies.**
(XLSX)

**S6 Table. Additional characteristics of included studies.**
(DOCX)

**S7 Table. Detailed risk of bias (RoB 2 and ROBINS-I).**
(XLSX)

**S8 Table. Quality of evidence (GRADE).**
(DOCX)

**S9 Table. Raw data of included studies in further analysis.**
(XLSX)

**S10 Figures. Content and composition design patterns, dashboard evaluation.**
(DOCX)

## Acknowledgments

We thank Dr. Giuseppe Pichierri for his contribution to the design and creation of the figures.

## Author contributions

**Conceptualization:** Stefan Markun, Jakob M. Burgstaller.

**Data curation:** Florence L. Meier, Levy Jäger, Jakob M. Burgstaller.

**Formal analysis:** Levy Jäger, Jakob M. Burgstaller.

**Investigation:** Florence L. Meier, Stefan Markun.

**Methodology:** Jakob M. Burgstaller.

**Project administration:** Florence L. Meier, Jakob M. Burgstaller.

**Software:** Levy Jäger.

**Supervision:** Jakob M. Burgstaller.

**Validation:** Florence L. Meier, Jakob M. Burgstaller.

**Visualization:** Levy Jäger.

**Writing – original draft:** Florence L. Meier, Jakob M. Burgstaller.

**Writing – review & editing:** Florence L. Meier, Levy Jäger, Oliver Senn, Stefan Markun, Jakob M. Burgstaller.

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
