## [Decision Letter · Decision Letter 0]

Dear Dr. Burgstaller,

Thank you for submitting your manuscript to PLOS ONE. After careful consideration, we feel that it has merit but does not fully meet PLOS ONE’s publication criteria as it currently stands. Therefore, we invite you to submit a revised version of the manuscript that addresses the points raised during the review process.

**ACADEMIC EDITOR: **
**Thank you for submitting your article to PLOSONE. Please address the comments made by the two reviewers. They have provided some valuable recommendations to consider. **

We look forward to receiving your revised manuscript.

Kind regards,

Niklas Bobrovitz

Academic Editor

PLOS ONE

Journal Requirements:

2. As required by our policy on Data Availability, please ensure your manuscript or supplementary information includes the following:

3. Please remove all personal information, ensure that the data shared are in accordance with participant consent, and re-upload a fully anonymized data set.

Reviewers' comments:

Reviewer's Responses to Questions

**Comments to the Author**

1. Is the manuscript technically sound, and do the data support the conclusions?

Reviewer #1: Yes

Reviewer #2: Yes

2. Has the statistical analysis been performed appropriately and rigorously?

Reviewer #1: N/A

Reviewer #2: N/A

3. Have the authors made all data underlying the findings in their manuscript fully available?

Reviewer #1: Yes

Reviewer #2: Yes

4. Is the manuscript presented in an intelligible fashion and written in standard English?

Reviewer #1: Yes

Reviewer #2: Yes

Reviewer #1: This paper describes the findings of a systematic review of the effectiveness of dashboards in primary care, especially in the area of audit and feedback. It describes the process and methodology very well and the results and discussion sections are clear.

I have some minor revision suggestions:

Line 179: change ‘One studies’ to ‘One study’.

Table 2: change ‘Primary Outcome(s)’ and ‘Secondary Outcome(s)’ to ‘Primary Target(s)’ and ‘Secondary Target(s)’ to clearly differentiate these from Primary / Secondary Outcomes Results.

Line 268: change ‘screenshots of the dashboards, what made the’ into ‘screenshots of the dashboards, which made the’ (it should be which, not what).

Line 277: What do you mean by ‘detailed visualizations’? Please give examples.

Line 278: Why do you call gauges and arrows ‘advanced visual elements’? Please explain or add a reference.

Line 280: Please briefly explain ‘open, table and grouped’.

Line 297: The sentence starting ‘Additionally, four studies collected mainly positive subjective feedback …’ is confusing. I presume that the feedback was positive but it could also mean that they only collected positive feedback and disregarded anything negative. Please make this clearer. Also, remove the word ‘Additionally’.

Reviewer #2: Dear Editors and Authors,

Thank you for the opportunity to review this timely and well-conceived systematic review. The manuscript addresses a highly relevant topic within the evolving field of digital health—namely, the use of interactive dashboards as audit and feedback (A&F) tools to improve care quality in primary care settings. The study is methodologically sound, well-articulated, and offers a valuable synthesis of current evidence. It aligns with the standards of PRISMA 2020, is registered on PROSPERO (CRD42024506727), and applies appropriate tools for risk of bias (RoB 2, ROBINS-I) and quality of evidence assessment (GRADE).

Despite its strengths, the manuscript would benefit from minor revisions to expand its discussion and address certain critical domains that were underexplored. These additions would enhance the paper’s conceptual rigor, applicability in real-world settings, and alignment with current discussions in implementation science and digital health policy.

1. Strengths of the Manuscript

• Clearly formulated research question with operational definitions of interactive A&F dashboards.

• Broad and sensitive search strategy across five databases, executed by an information specialist.

• Transparent screening, data extraction, and risk of bias assessment by independent reviewers.

• Appropriately justified narrative synthesis in light of heterogeneity among studies.

• Well-structured manuscript with fluent and professional English writing.

• Full data availability in accordance with PLOS ONE policies.

2. Areas for Enhancement

2.1. Data Security, Privacy, and Anonymization

The review does not address the data protection and privacy implications of using interactive dashboards in clinical settings—particularly given their reliance on electronic health record (EHR) data. There is no discussion of anonymization protocols or compliance with legal and ethical frameworks such as GDPR, HIPAA, or other relevant national data governance policies. This is a notable omission, especially considering the increasing scrutiny around patient data handling in digital health solutions.

2.2. Data Input Quality and Information Reliability

The accuracy of any dashboard is only as reliable as the quality of the data it receives. The authors do not discuss the risk of poor data quality—such as incomplete, outdated, or incorrectly coded clinical records—compromising the validity of the dashboards' outputs. This is a critical limitation that should be explicitly acknowledged and discussed.

2.3. Digital and Health Literacy of End-Users

The successful implementation of dashboards is contingent on clinicians’ ability to interpret visualizations and engage with interactive digital tools. The manuscript does not consider the heterogeneity in digital and health literacy among primary care professionals, which may act as a barrier to uptake and utilization.

2.4. Digital Inclusion and Infrastructure Readiness

The included studies are drawn from high-income settings, but the manuscript does not consider variability in infrastructure—such as internet connectivity, hardware availability, and IT support—which may significantly affect dashboard implementation in low-resource environments. These contextual factors should be discussed to better inform the generalizability of the findings.

2.5. Interoperability Across Information Systems

A major omission in the discussion is the lack of attention to interoperability challenges. Many health systems operate with diverse, non-integrated platforms. The feasibility and impact of dashboards are critically dependent on their ability to interoperate with existing EHRs and data repositories. Without such integration, dashboards may become redundant or underutilized. This topic is absent from the current version and should be included.

2.6. Reviewer Concordance (Kappa Statistic)

While the manuscript notes that two independent reviewers conducted the screening process, no measure of interrater agreement (e.g., Cohen’s kappa) is reported. Although optional under PRISMA, including a concordance metric is a recommended good practice to demonstrate methodological rigor.

3. Recommendations for Revision

I encourage the authors to:

• Add a paragraph in the Discussion section explicitly addressing the above issues, especially data privacy, system interoperability, and infrastructure gaps.

• Acknowledge these limitations in the “Implications for Practice” or “Limitations” section, emphasizing the importance of digital readiness and equity.

• Where possible, clarify whether any of the included studies reported on actual dashboard usage analytics, user feedback, or usability testing—information that would be valuable for implementation stakeholders.

• Consider reporting (or explicitly noting the absence of) interrater agreement metrics during study selection.

4. Final Verdict: MINOR REVISION

This manuscript offers a valuable and methodologically sound contribution to the literature on digital quality improvement tools in primary care. The findings are relevant to practitioners, researchers, and policymakers. With the inclusion of the recommended revisions—particularly those expanding the scope of the discussion—the article will be significantly strengthened in both academic and practical terms.

Recommendation: Minor Revision

Thank you once again for the opportunity to engage with this work.

Sincerely,

Dr. André Ramalho

PhD in Public Health

Corporative Health Manager

Health Systems Researcher | Digital Health and Policy Advisor

**Do you want your identity to be public for this peer review?** For information about this choice, including consent withdrawal, please see our Privacy Policy

Reviewer #1: **Yes: ** Heike Vornhagen

Reviewer #2: **Yes: ** Andre Ramalho, PhD

---

## [Author Response · Author response to Decision Letter 1]

4 Jun 2025

Comments to the reviews on the manuscript (see also separate uploaded file)

Effectiveness of interactive dashboards as audit and feedback tools in primary care: a systematic review

June 4, 2025

We thank the reviewers and the editor for their very valuable thoughts and comments. We have improved our manuscript accordingly and now believe to provide balanced clinically relevant information for clinicians and researchers. If any other changes are needed, we are happy to include additional aspects in the manuscript.

Journal Requirements:

1

Please ensure that your manuscript meets PLOS ONE's style requirements, including those for file naming. The PLOS ONE style templates can be found at https://journals.plos.org/plosone/s/file?id=wjVg/PLOSOne_formatting_sample_main_body.pdf and https://journals.plos.org/plosone/s/file?id=ba62/PLOSOne_formatting_sample_title_authors_affiliations.pdf

Answer 1:

We checked the requirements.

2

As required by our policy on Data Availability, please ensure your manuscript or supplementary information includes the following:

Answer 2:

See 2a to 2g

2a

Answer 2a:

S5 Table (divided into full-text and title/abstract exclusion sheet)

2b

Answer 2b:

S5 Table Column D in each sheet

2c

Answer 2c:

Not applicable.

2d

Answer 2d:

S9 Table

2d1

Answer 2d1:

S9 Table Columns B & C

2d2

Answer 2d2:

S9 Table Column D

2d3

Answer 2d3:

S9 Table Columns E to M

2e

Answer 2e:

Not applicable

2f

Answer 2f:

S7 Table for detailed risk of bias. S8 Table for detailed quality of evidence (GRADE).

2g

Answer 2g:

In the methods section on lines 83 to 84 it reads:

“Further, reference lists of included studies and reviews were examined to identify additional studies.”

In the limitations on lines 400 to 402 it reads:

“Despite these challenges, we are confident that our sensitive search strategy, coupled with manual screening of potentially eligible studies cited in the included articles and systematic reviews, minimized the likelihood of missing relevant studies.”

3

Please remove all personal information, ensure that the data shared are in accordance with participant consent, and re-upload a fully anonymized data set.

Answer 3:

We checked the requirements.

4

Please include captions for your Supporting Information files at the end of your manuscript, and update any in-text citations to match accordingly. Please see our Supporting Information guidelines for more information: http://journals.plos.org/plosone/s/supporting-information.

Answer 4:

We added a Supporting Information content at the end of the manuscript according to the journal guidelines.

5

Answer 5:

We checked the reference list.

Reply to Reviewer #1:

1

This paper describes the findings of a systematic review of the effectiveness of dashboards in primary care, especially in the area of audit and feedback. It describes the process and methodology very well and the results and discussion sections are clear.

Answer 1:

We thank the reviewer for their very encouraging compliments.

2

Line 179: change ‘One studies’ to ‘One study’.

Answer 2:

Thank you for pointing out the spelling mistake.

3

Table 2: change ‘Primary Outcome(s)’ and ‘Secondary Outcome(s)’ to ‘Primary Target(s)’ and ‘Secondary Target(s)’ to clearly differentiate these from Primary / Secondary Outcomes Results.

Answer 3:

We changed the column headers accordingly.

4

Line 268: change ‘screenshots of the dashboards, what made the’ into ‘screenshots of the dashboards, which made the’ (it should be which, not what).

Answer 4:

Thank you for pointing out grammatical error.

5

Line 277: What do you mean by ‘detailed visualizations’? Please give examples.

Answer 5:

Thank you for your comment. Bach et el. (2023) defined detailed visualizations as follows (paraphrased):

“Detailed visualizations are comprehensive visual components within a dashboard that provide a high level of data detail and precision, suitable for viewers who need to read exact values or explore the data in depth.”

We interpreted visualizations as 'detailed visualizations' when they integrated graphs (e.g., bar graphs or time-series plots) together with additional contextual information (combined visual components) - such as numeric annotations, clear axis labeling, informative legends, and supplementary metrics - embedded within dashboards, providing sufficient detail to enable users to accurately read and interpret precise data values.

We added the following sentence to the manuscript on lines 282 to 283:

“[…](combined visual components including both graphs and additional informational elements) [… ]”

Unfortunately, the definition was lost in the S3 Table, therefore we added it there. Further, we added to the manuscript the following information (see also answer #7):

Lines 273 and 285: “[…] for definitions see S3 Table)”

Line 294: “[…] for definitions see S4 Table)”

6

Line 278: Why do you call gauges and arrows ‘advanced visual elements’? Please explain or add a reference.

Answer 6:

Thank you for your comment. Unfortunately, we chose an inappropriate wording. We replaced “advanced” with “further”.

7

Line 280: Please briefly explain ‘open, table and grouped’.

Answer 7:

In S3 Table, each dashboard design pattern is listed with a definition. We added on line 273 and 285 the following reference “for definitions see S3 Table”.

To keep the manuscript concise, we refrain from adding these definitions into the result text. Please find below the definitions extracted from S3 Table:

Open Layouts: Widgets are placed without specific alignment rules, often in a grid.

Table Layouts: Widgets are organized into rows and columns for easy information retrieval.

Grouped Layouts: Widgets are grouped based on a specific relation, often labeled by a common title.

8

Line 297: The sentence starting ‘Additionally, four studies collected mainly positive subjective feedback …’ is confusing. I presume that the feedback was positive but it could also mean that they only collected positive feedback and disregarded anything negative. Please make this clearer. Also, remove the word ‘Additionally’.

Answer 8:

Thank you for your valuable comment. We adjusted the sentence accordingly, and it now reads as follows (lines 303 to 306):

“Four studies reported that most of the subjective feedback collected on the i-A&F dashboard’s usability and user satisfaction (“perceived engagement”) was positive. This feedback was gathered through either user involvement during development [22, 23] or post-hoc interviews [24, 25].”

Reply to Reviewer #2:

1

Strengths of the Manuscript

• Clearly formulated research question with operational definitions of interactive A&F dashboards.

• Broad and sensitive search strategy across five databases, executed by an information specialist.

• Transparent screening, data extraction, and risk of bias assessment by independent reviewers.

• Appropriately justified narrative synthesis in light of heterogeneity among studies.

• Well-structured manuscript with fluent and professional English writing.

• Full data availability in accordance with PLOS ONE policies.

Answer 1:

Thank you for highlighting the strengths of our review.

2

Areas for Enhancement

Answer 2:

We have carefully considered each point and integrated appropriate clarifications and limitations within the manuscript. However, several suggested expansions lie beyond our explicitly stated research scope—specifically, evaluating the clinical effectiveness of interactive dashboards as audit and feedback tools within primary care settings.

Nevertheless, to transparently address the points raised, we have explicitly noted relevant limitations in our revised manuscript, highlighting important aspects such as data privacy and user literacy as critical considerations for future research in digital health implementation.

2.1

Data Security, Privacy, and Anonymization

The review does not address the data protection and privacy implications of using interactive dashboards in clinical settings—particularly given their reliance on electronic health record (EHR) data. There is no discussion of anonymization protocols or compliance with legal and ethical frameworks such as GDPR, HIPAA, or other relevant national data governance policies. This is a notable omission, especially considering the increasing scrutiny around patient data handling in digital health solutions.

Answer 2.1:

We acknowledge that data security, privacy implications, and adherence to frameworks such as GDPR and HIPAA are critical aspects of implementing digital health solutions. The included studies don’t provide a discussion on the broader data protection and privacy implications of implementing or using interactive dashboards. Further, these aspects were beyond the scope of this systematic review, which focused explicitly on the effectiveness of interactive dashboards. Future research should include an explicit assessment of compliance and best practices in data handling (see 3.1 below).

2.2

Data Input Quality and Information Reliability

The accuracy of any dashboard is only as reliable as the quality of the data it receives. The authors do not discuss the risk of poor data quality—such as incomplete, outdated, or incorrectly coded clinical records—compromising the validity of the dashboards' outputs. This is a critical limitation that should be explicitly acknowledged and discussed.

Answer 2.2:

While the quality of data input into dashboards undeniably influences their utility and accuracy, assessing data quality and reliability was beyond this review’s scope.

A brief review revealed that some studies affirm that steps were taken to ensure reliability (Jones 2023), refer to previous validation efforts for key data points (Linder 2010), acknowledge how data entry methods affect data capture (Jones 2023), and link general data quality improvements to policy contexts (de Lusignan 2021). Therefore, future research should specifically evaluate the impact of data quality on dashboard effectiveness (see 3.1 below).

2.3

Digital and Health Literacy of End-Users

The successful implementation of dashboards is contingent on clinicians’ ability to interpret visualizations and engage with interactive digital tools. The manuscript does not consider the heterogeneity in digital and health literacy among primary care professionals, which may act as a barrier to uptake and utilization.

Answer 2.3:

We acknowledge that variability in digital and health literacy among primary care clinicians may significantly influence dashboard uptake and effectiveness. However, our review did not address this aspect explicitly, as it aimed solely at synthesizing available evidence on clinical effectiveness.

A brief review revealed that the sources do not explicitly label clinician digital or health literacy heterogeneity as a barrier using those precise terms, but they strongly suggest that factors like inadequate training and support (Linder 2010, Guldberg 2011), poor tool usability and integration (Guldberg 2011, Jones 2023), and potentially a lack of understanding regarding underlying data capture methods (Jones 2023) significantly impede the successful uptake and utilization of dashboard interventions by primary care professionals. The inclusion of training, support, and user-centric design (like co-design) in some interventions suggests an implicit recognition that user capabilities and comfort with technology are critical for success. We added the following sentence to the “Implications for practice and future research” section in the discussion (lines 395 to 396):

“[…], particularly given the heterogeneity in digital and health literacy among clinicians, which may influence uptake and successful utilization.”

2.4

Digital Inclusion and Infrastructure Readiness

The included studies are drawn from high-income settings, but the manuscript does not consider variability in infrastructure—such as internet connectivity, hardware availability, and IT support—which may significantly affect dashboard implementation in low-resource environments. These contextual factors should be discussed to better inform the generalizability of the findings.

Answer 2.4:

Our systematic review primarily included studies from high-income settings, limiting the generalizability of findings to low-resource environments, where infrastructure readiness may vary significantly. We agree that future reviews and primary studies should explicitly consider digital inclusion and infrastructure readiness to ensure broader applicability (see 3.1 below).

2.5

Interoperability Across Information Systems

A major omission in the discussion is the lack of attention to interoperability challenges. Many health systems operate with diverse, non-integrated platforms. The feasibility and impact of dashboards are critically dependent on their ability to interoperate with existing EHRs and data repositories. Without such integration, dashboards may become redundant or underutilized. This topic is absent from the current version and should be included.

Answer 2.5:

Interoperability of dashboards with existing EHR systems is critical for practical implementation and scalability. We cover this aspect with the dashboard evaluation scenario “system implementation” in the result section. We expanded the corresponding part and it now reads as follows (lines 307 to 308):

“Only two i-A&F dashboards [23, 24] were directly integrated into existing EHRs. The other four i-A&F dashboards [22, 25-27] were independent of the EHR (“system implementation”).”

Nonetheless, future res

---

## [Decision Letter · Decision Letter 1]

Effectiveness of interactive dashboards as audit and feedback tools in primary care: a systematic review

PONE-D-25-01595R1

Dear Dr. Burgstaller,

We’re pleased to inform you that your manuscript has been judged scientifically suitable for publication and will be formally accepted for publication once it meets all outstanding technical requirements.

Kind regards,

Niklas Bobrovitz

Academic Editor

PLOS ONE

Additional Editor Comments (optional):

Reviewers' comments:

Reviewer's Responses to Questions

**Comments to the Author**

Reviewer #1: All comments have been addressed

Reviewer #2: All comments have been addressed

2. Is the manuscript technically sound, and do the data support the conclusions?

Reviewer #1: (No Response)

Reviewer #2: Yes

3. Has the statistical analysis been performed appropriately and rigorously?

Reviewer #1: (No Response)

Reviewer #2: N/A

4. Have the authors made all data underlying the findings in their manuscript fully available?

Reviewer #1: (No Response)

Reviewer #2: Yes

5. Is the manuscript presented in an intelligible fashion and written in standard English?

Reviewer #1: (No Response)

Reviewer #2: Yes

Reviewer #1: (No Response)

Reviewer #2: Congratulations to the authors for the article. I hope that they will soon continue to address the points I made in future studies. Since the authors have addressed all of my concerns, I recommend that it be published.

**Do you want your identity to be public for this peer review?** For information about this choice, including consent withdrawal, please see our Privacy Policy

Reviewer #1: **Yes: ** Heike Vornhagen

Reviewer #2: **Yes: ** Prof. Dr. Andre Luis C Ramalho, PhD

---

## [Editor Report · Acceptance letter]

PONE-D-25-01595R1

PLOS ONE

Dear Dr. Burgstaller,

I'm pleased to inform you that your manuscript has been deemed suitable for publication in PLOS ONE. Congratulations! Your manuscript is now being handed over to our production team.

Kind regards,

on behalf of

Dr. Niklas Bobrovitz

Academic Editor

PLOS ONE